# Amplified Vasodilatation within the Referred Pain Zone of Trigger Points Is Characteristic of Gluteal Syndrome—A Type of Nociplastic Pain Mimicking Sciatica

**DOI:** 10.3390/jcm10215146

**Published:** 2021-11-02

**Authors:** Elzbieta Skorupska, Tomasz Dybek, Michał Rychlik, Marta Jokiel, Jarosław Zawadziński, Paweł Dobrakowski

**Affiliations:** 1Department of Physiotherapy, Poznan University of Medical Sciences, 61-701 Poznan, Poland; marta.jokiel@gmail.com (M.J.); jaroslaw.zawadzinski@gmail.com (J.Z.); 2Faculty of Physical Education and Physiotherapy, Opole University of Technology, 45-758 Opole, Poland; dtdybek@gmail.com; 3Department of Virtual Engineering, Poznan University of Technology, 60-965 Poznan, Poland; rychlik.michal@poczta.fm; 4Department of Traumatology, Orthopedics and Hand Surgery, Poznan University of Medical Sciences, 60-761 Poznan, Poland; 5Psychology Institute, Humanitas University in Sosnowiec, 41-200 Sosnowiec, Poland; paweldobrakowski@interia.pl

**Keywords:** pain measurement, thermography, physical stimulation, skin temperature, MATLAB

## Abstract

Gluteal syndrome (GS) mimicking sciatica is a new disease that has been recently recognized and included in the International Classification of Diseases, 11th Revision. The present study examines nociplastic pain involvement in GS and sciatica patients using a new Skorupska protocol (SP) test that provokes amplified vasodilatation in the area of expected muscle-referred pain. A positive test is confirmed if there is (i) a development of autonomic referred pain (AURP) and (ii) an increase in the delta of average temperature (Δ₸°) > 0.3 °C at the end of the stimulation and during the observation SP phases. Chronic GS (*n* = 20) and sciatica (*n* = 30) patients were examined. The SP test confirmed muscle-referred pain for (i) all GS patients with 90.6% positive thermograms (Δ₸° 0.6 ± 0.8 °C; maximum AURP 8.9 ± 13.6% (both *p* < 0.05)) and (ii) those sciatica (*n* = 8) patients who reported pain sensation during the test with 20.6% positive thermograms (Δ₸° 0.7 ± 0.7 °C; maximum AURP 15.1 ± 17.8% (both *p* < 0.05)). The remaining sciatica (*n* = 22) patients did not report pain during the test and presented a Δ₸° decrease and the AURP size below 1%. Conclusion: Amplified vasodilatation suggesting nociplastic pain involvement was confirmed for all GS and sciatica patients who reported painful sensations in the zone typical for gluteus minimus referred pain during the test.

## 1. Introduction

The newest data indicate that low back pain (LBP) is characterized by different types of pain, including nociceptive pain, neuropathic (radicular) pain that travels down the legs, and—in some cases—nociplastic pain, which is caused by the amplification of pain in the central nervous system. It has been noted that these pain subtypes can overlap. For example, a patient may present with sciatica and diffuse symptoms outside of the usual pathoanatomical referral patterns [1,2]. A new subtype of LBP involving leg pain, known as gluteal syndrome (GS), was included for the first time in the 11th Revision of the International Classification of Diseases [3]. GS is an example of nociplastic pain that can mimic sciatica or that can overlap with real sciatica. GS is characterized as a local syndrome of the lower limbs that provokes myofascial pain from trigger points located in one of the three gluteal muscles (i.e., gluteus maximus, medius, or minimus) [3]. Originally, a trigger point was defined as “a hyperirritable spot within a taut band of a skeletal muscle that is painful on compression, stretch, overload, or contraction of the tissue which usually responds with a referred pain that is perceived distant from the spot” [4]. The presence of H+, BK, CGRP, SP, TNF-, IL-1, serotonin, and norepinephrine has been confirmed to be in the vicinity of trigger points (TrPs) [5]. However, Travell and Simons’ clinical criteria, which are often used to confirm the presence of TrPs, have provoked doubts and controversies. Recently, a revision of the criteria was proposed, and the necessity for referred-pain confirmation was underlined [6]. It is not widely known that muscle-referred pain is a pain sensation that occurs at a remote site far from the linked trigger points and is considered an important marker of the central sensitization involvement in the pathomechanisms of TrPs. It was previously noted that sign pressure over these TrPs provokes a referred pain pattern that is recognized by patients as “sciatic pain” and that a straight leg raise (SLR) test is usually restricted due to tightness in the hamstring and gluteus maximus muscles [3]. Although trigger points have been legitimized in medicine by ICD-11, controversies remain due to a lack of objective confirmation for TrPs [7]. Indeed, the varied and complicated etiology of low-back leg pain still causes disagreement among researchers [8]. There is consensus that it is impossible to objectively distinguish the lumbosacral radicular syndrome defined as sciatica from other pain syndromes based on standard neurological examinations and magnetic resonance imaging [9,10,11,12,13,14]. The importance of the autonomic nervous system (ANS) for chronic low back pain has been highlighted [15,16,17]. A new validated method—the Skorupska protocol (SP) test—is a test that is intended to confirm muscle-referred pain (nociplastic pain related to muscle) based on a transient, amplified vasomotor phenomenon that possibly occurs due to ANS activity within the patient’s daily complaint area [18,19,20]. The SP test is positive if it a simultaneous increase in both diagnostic parameters is observed with the observed ROI, namely (i) delta changes in the average temperature (Δ₸°) of more than 0.3 °C and (ii) the development of autonomic referred pain (AURP).

This study examines the involvement of nociplastic pain in GS and sciatica patients based on the presence of amplified vasodilatation reflecting muscle-referred pain.

## 2. Material and Methods

### 2.1. Study Design

This study was approved by the Ethics Committee of the Poznan University of Medical Sciences (number 689/20) and was conducted in accordance with the Declaration of Helsinki. Before data collection, all subjects gave written informed consent to participate in the study. A detailed description of all of the examinations and treatment procedures, including muscle noxious stimulation as well as the risks involved in the study, was provided to the participants. The participants had the right to refuse the SP test and withdraw from the study at any time without penalty.

Gluteal syndrome (GS) is a subtype of chronic low-back leg pain that mimics sciatica. GS is defined as aching myofascial pain arising from trigger points located in one of the three gluteal muscles. The GS diagnostic criteria are as follows: (i) sign pressure on the trigger point that is responsible for the pain occurrence and that reproduces the referred pain pattern; for gluteus minimus—the pain pattern is similar to that of sciatica, whereas for the gluteus medius and gluteus maximus—pain is referred to the buttock and rarely to the thigh; (ii) straight-leg raising is usually restricted because of the tightness in the hamstring and gluteus maximus muscles. GS is classified as nociplastic pain that is related to the central sensitization process (nociplastic pain) and that is based on the presence of primary hyperalgesia (trigger points presence) and secondary hyperalgesia (muscle-referred pain). 

The SP test confirms nociplastic pain if a patient with trigger points presents test-provoked vasodilatation and/or vasoconstriction that is dependent on the tested muscle in the muscle-referred zone (positive test, SP(+)). The validation study confirmed that a positive SP test resulted in the autonomic phenomenon for trigger points exclusively. The localization of the autonomic phenomenon was coincident with both the pain sensation reported by the patients during the test and their daily complaint area. TrPs-negative pain patients and healthy subjects did not present any autonomic phenomenon (negative test, SP(−)), and they reported no pain sensation during the test. For gluteus minimus trigger points, the thigh was confirmed as the diagnostic area of the SP test [20].

A secondary analysis of the thermal data extracted from previously published studies was performed to determine the differential diagnosis of the GS and sciatica subjects based on the SP test [18,21,22]. Secondly, we analyzed the thermal data of the chronic sciatica patients who reported pain sensations provoked by the SP test. All of the GS patients had gluteal trigger points and reported pain sensations recognized as the daily complaint during the SP test, which means that they were SP(+). The sciatica patients without TrPs who reported pain sensations recognized as the daily complaint during the SP test were classified as having SP(+) sciatica. The sciatica patients without TrPs who did not report pain sensations during the SP test were classified as having SP(−) sciatica.

The current study includes 95% unpublished thermal data. Our previous studies presented 3 representative thermograms out of 320 thermograms recorded during the 16 min procedure.

### 2.2. Participant Characteristics

Eighty-five patients were assessed for eligibility. Thirty-five (*n* = 35) of them were excluded: eighteen (*n* = 18) for not meeting the inclusion criteria and seventeen (*n* = 17) declined to participate. All of the participants finished the diagnostic protocol. The examined patients presented variable leg-pain zones (thigh, calf, and foot); only the thigh zone was painful for every participant (Table 1). The inclusion criteria for participants were age between 30 and 60 and unilateral lower leg pain >3 on a 1–10 point visual analog scale (VAS). Additionally, for gluteal syndrome, the inclusion criteria were as follows: diagnosis of active trigger points within the gluteus minimus muscle, a chronic state, and a positive SLR test (>45°). The inclusion criteria for sciatica were as follows: diagnosis of sciatica based on a neurological examination, a positive SLR test (>30° and <60°), a herniated disc with nerve-root compression confirmed by magnetic resonance imaging (MRI), a lack of trigger points within the gluteus muscles, and a lack of reproducible lower leg symptoms experienced by patients and that is recognized as familiar due to the deep snapping palpation of the gluteus minimus muscle. Patients were excluded from the study if they presented with previous back surgery, pregnancy, cauda equina syndrome, complex regional pain syndrome, spinal tumors, scoliosis, coagulant treatment, disseminated intravascular coagulation, diabetes, stroke, epilepsy, infection, inflammatory rheumatologic diseases, or oncological history. 

### 2.3. Method

Every patient classified as having gluteal syndrome (*n* = 20) or chronic sciatica (*n* = 30) was diagnosed by an experienced neurologist and a myofascial pain specialist with over 10 years of experience. Then, each of the participants was assessed using the SP test (detailed test description below). Gluteal syndrome was confirmed if a patient presented a minimum of two active trigger points within the gluteus minimus muscle according to Travell and Simons’ diagnostic criteria—i.e., a taut band, spot tenderness, pain recognition, and a limited range of movement. Moreover, following the newest recommendations, we considered additional diagnostic criteria, such as the reproduction of the lower leg symptoms experienced by the patient and that were recognized as familiar, the rarely accessible taut band criterion, and spontaneous referred pain [6]. Next, each patient underwent a functional examination that included a straight leg raise test. In the current study, the GS diagnosis was supported by positive results at the validated points of measurement (the end of the stimulation phase and the end of the observation phase). The diagnosis of sciatica (*n* = 30) of radicular origin was based on a clinical bedside examination accompanied by a positive straight leg raise test and MRI results. 

#### SP Test

A positive test was confirmed if a patient with trigger points presented test-provoked vasodilatation and/or vasoconstriction in the daily complaint area that was coincident with the region of the referred pain pattern of the tested muscle. The autonomic phenomenon is defined by two parameters. The first parameter is the percentage size of the vasomotor reactivity subarea with a temperature that is not registered for the patient before the stimulation (T0), which was named autonomic referred pain (AURP_T0_). The second one is an increase in the delta of the average temperature (Δ₸°) in the examined body part of more than 0.3 °C. Both diagnostic parameters were demanded to classify the test results as positive (SP(+)).

Detailed AURP_T0_ explanation: 

Autonomic referred pain (AURP_T0_) reflects the size of the phenomenon revealing the temperature that was not registered for the patient before the stimulation. Technically, AURP_T0_ is an isolated region (segmented part of the thermogram) with a defined temperature range that is individually measured in the ROI for every 3 s of the procedure for every case. Originally, the AURP_T0_ size (cm^2^) was calculated manually by a technician, and then the percentage size of the area within the thigh, calf, and foot was calculated separately. That method allowed the size of the AURP to be calculated at the end of both test phases. Currently, the new codes created for the purposes of the test in the MATLAB environment allowed an automated AURP size calculation for each of the 320 recorded thermograms [18]. 

Description of the exploration:

Both full and truncated descriptions of the method were presented in our previous studies [18,19]. The description of the MATLAB procedure for the SP test, which allowed the analysis of the test results every 3 s, is provided in the Appendix A.

Short description of the exploration:
(1)Trigger point examination based on the palpatory diagnostic criteria according to Travell and Simons [23]. (2)Thermal side-to-side comparison of the patients at rest to exclude a possible neuropathic pain component (confirmed there was a temperature decrease of more than 0.5 °C if in the pain subarea compared to the opposite side).(3)Examination towards the autonomic response of the trigger points at the daily complaint area that is coincident with the referred pain zone for the examined muscle.(4)Numerical analysis of the thermograms.


The examination consisted of two parts: (i) noxious, nociceptive muscle stimulation under thermal camera control of the area with expected muscle-referred pain (10 min) and (ii) the post-stimulation resting phase—further thermal observation of patients at rest (6 min). For part (i), the noxious stimulation of the two most sensitive trigger points (GS) or the most tender muscle areas (sciatica) within the gluteus minimus muscle was applied using the continuous fast-in-fast-out dry needling technique. During the whole noxious stimulation, the patients reported the localization of the pain sensation provoked by needling (thigh, calf, foot).

Numerical analysis of the thermograms: A comparative analysis of the thermograms (320) was registered every 3 s, and the thermogram recorded for the patient before the stimulation was performed using MATLAB. For every thermogram, the thigh subarea was segmented. Then, both the AURP_T0_ size and Δ₸° of the thigh region were calculated. The significance of both diagnostic parameters was checked for each of the 320 thermograms.

An illustration of the SP test and an example of the test results are shown in Figure 1.

### 2.4. Sample Size Calculation

The sample size for this publication was limited due to the patient availability for each group. Therefore, non-parametric tests were performed instead of ANOVA. For an exact test at a significance level of 0.05 with a beta power of 0.95, the lowest possible sample size is *n* = 19. The total sample size for this publication was *n* = 50.

### 2.5. Statistical Analysis

Exact two-tailed Mann–Whitney U tests with corrected ties were performed to assess the differences between gluteal syndrome (*n* = 20) and sciatica (*n* = 30) patients depending on the noxious stimulation sensitivity reported during the SP test. The tests were applied to compare the data for AURP and Δ₸° between the aforementioned groups. To obtain the significance of the *p* values, a post hoc Dunn test was performed. Due to the multiple comparison problem, the aforementioned test was corrected using the Holm–Sidak procedure. The Dunn test was prepared using the dunn.test package in R. Values, figures, and tables in the text are expressed as the means ± standard deviation (SD) or as quartiles with the median. The significance level was set for all tests are at *p*  <  0.05. Statistical analysis was performed using IBM SPSS Statistics version 26 and MATLAB version R2021. Firth’s Regression was prepared in R version 4.0.5 using the logistf package (IBM Corp. Released 2019. IBM SPSS Statistics for Windows, Version 26.0. Armonk, NY, USA: IBM Corp.).

## 3. Results

The SP test confirmed muscle-referred pain based on the presence of amplified vasodilatation among the gluteal syndrome and sciatica patients who reported pain provocation in the symptomatic region (SP(+)). Both groups presented amplified vasodilatation in the expected diagnostic zone (thigh ROIs), which differentiated them significantly from the sciatica patients who did not present any positive response to the test (SP(−)) (*p* < 0.05). The AURP_T0_ and Δ₸° trends over time and the significant differences for both (*) parameters together, which were measured every 3 s, are shown in Figure 2. The main characteristics of the SP test results for the examined groups are shown in Table 2. A video presenting the test response of a GS and sciatica SP(−) case is available as Appendix A.

### A Detailed Description of the SP Test Parameters Depending on the Disease and Test Phase

A summary of the statistically significant SP test results for every patient subgroup depending on the test phase is shown in Table 3. 

## 4. Discussion

The results of the study confirmed the presence of amplified vasodilatation within the daily complaint area of every gluteal syndrome patient and among chronic sciatica patients who had no trigger points but who reported pain sensations during noxious muscle stimulation for the first time (Figure 2). Most of the GS thermograms were test-positive (Table 2). The SP(+) sciatica patients presented an autonomic phenomenon in one out of four thermograms (*p* < 0.05), with a bigger AURP_T0_ size compared to the GS patients (Table 2). The analysis of all of the registered thermograms revealed the fluctuating characteristic of the observed phenomenon during the stimulation phase followed by further vasodilatation development during the observation phase of the SP test (Figure 2) The AURP size seemed to be more sensitive to noxious stimulation than Δ₸° and varied significantly starting from the first few minutes of the procedure (Appendix A). Dry needling of the trigger points (GS) or tender muscle areas (sciatica) was the noxious stimulant used during the SP test. This method can provoke a twitch response (TW), which can be defined as a probable autonomic spinal cord reflex resulting in a brisk muscle contraction followed by a short-lasting referred pain pattern. AURP may have been provoked by a twitch response caused by a spinal reflex following mechanical needle stimulation [24,25]. Furthermore, the quantity of brisk muscle fiber contractions within the taut band used for the TW description was previously linked to the irritability of TrPs [26]. Based on the physiological background of the observed phenomenon, it can be assumed that the SP test provoked a temporarily amplified ANS imbalance in the area of referred pain induced by a TW series followed by referred pain, which was possibly reflected in the fluctuating AURP reactivity in the first few minutes of the noxious stimulation (video file). Furthermore, it is likely that the increase in Δ₸° depended on the number of reactions evoked over time. We hypothesize that the autonomic phenomenon development that occurred during the stimulation reflected the temporal summation characteristic of the central sensitization phenomena. In the last decade, central sensitization was postulated as the phenomenon that was responsible for the development and/or maintenance of TrPs [27,28,29].

The GS patients presented a smaller AURP_T0_ size compared to the SP(+) sciatica subjects (Figure 2). Furthermore, the size of AURP_T0_ for the GS and SP(+) chronic sciatica patients was smaller compared to previously published data of subacute and chronic sciatica patients with the co-existence of TrPs [19,21,22]. The same characteristic was observed for the second SP test parameter, Δ₸°, which was almost 50% lower compared to the sciatica patients, who presented a Δ₸° of around 1.4 + 0.2 °C (*p* < 0.005). 

Recently, it has been reported that the stimulus intensity, rather than the pain severity or clinical state, determines how strongly the autonomic nervous system (ANS) responds to noxious stimulation in the pain region [30]. Based on these findings, it can be speculated that the greater number of positive thermograms would have greater clinical importance for the assessment of nociplastic pain involvement. Further studies considering clinical assessment are needed to determine an explanation. 

It is likely that the amplified vasodilatation that was observed occurred due to the activation of the non-noradrenergic vasodilator system, which affected the processes of reflex cutaneous vasoconstriction and vasodilatation [31,32] and reflected a temporary autonomic nervous system (ANS) imbalance within the daily complaint area. Only a few published studies have indicated the autonomic nervous system’s importance in chronic low back pain [15,16,17]. The autonomic responses of GS patients are interesting in the context of objective disease confirmation, particularly if both the trigger-point diagnostic criteria and the straight leg raise test interpretations are questioned [14,33]. The results are even more interesting if we consider the consensus that it is impossible to objectively distinguish lumbosacral radicular syndrome, which is defined as sciatica from other pain syndromes, using standard neurological examinations and magnetic resonance imaging [34]. Since the SP test is designed to objectively confirm the presence of referred pain, it can have very promising and important applications.

The hypothesis of why the autonomic phenomenon was confirmed for some of the sciatica patients who did not present trigger points relates to the fact that chronic pain is characterized by dynamic interactions between sensory and contextual processes in the brain (i.e., cognitive, emotional, and motivational), which are mediated by feed-forward and feedback processes [35]. Some of the SP(+) sciatica patients may not have been able to distinguish the sensations provoked by palpatory pressure during trigger point examination in their perceived pain areas for many complex reasons. To test gluteus minimus trigger points, we added another marker for the presence of TrPs to the typical diagnostic TrP protocol—referred pain. According to the literature, only around 50% of active TrPs co-occur with referred pain [6]. The same problem has been recognized for other deeply located muscles. It is possible that deeply located trigger points could not be reached by pressure palpation but were accessible for needling [23]. We can hypothesize that the SP(+) sciatica patients were TrPs-positive, but due to our inclusion criteria, they were overlooked. 

If this hypothesis is correct, the SP test will allow us to visibly distinguish between patients with positive referred pain and those with negative low-back leg pain. It is true that chronic pain may lead to the hyperarousal of the sympathetic nervous system [26]. However, it is unlikely that a chronic pain pathomechanism other than TrPs can provoke reactivity similar to that of the SP test. To determine whether SP-sensitive sciatica subjects are in fact hidden TrPs-positive patients who should be diagnosed with gluteal syndrome, a re-examination of such patients using another objective tool, such as quantitative sensory testing (QST), would be required. However, to our knowledge, there is only one study on trigger-point referred-pain measurements using the QST [9]. Further studies considering SP-sensitive sciatica patients are necessary.

Limitations of this study:

The present study only considered SLR-positive gluteal syndrome patients. Future studies considering all GS patients are necessary. Furthermore, a larger group of subjects is necessary to observe calf and foot responses to the SP test. To improve the quality of the study, a control group should be considered. Moreover, the therapist who performed the dry needling in the present study was not blinded to the trigger-point diagnosis, which could have biased the results to some extent.

Consequences for knowledge and future research:

The main point of interest in the nociplastic pain field is to find an objective way to diagnose pain. It seems that the SP test could serve as a valuable tool for diagnosing central sensitization subtypes that are related to trigger points. Additionally, the SP test is interesting for modern medical thermography that is focused on developing strict diagnostic protocols that are typical for active dynamic thermography, as this type of thermography can better reveal pathological processes thanks to its high-contrast pictures. The advantages of the SP test include a detailed analysis of the segmented parts of the thermograms and the need to confirm significant results for both diagnostic parameters. Moreover, the fact that the SP test results were supported by MATLAB indicates that this technique meets the highest standards in medical thermography. Future studies should explore the application of the SP test together with quantitative sensory testing of other muscles with trigger points.

**Summary:** Amplified vasodilatation suggesting nociplastic pain involvement was confirmed for all GS and sciatica patients who reported pain sensations in the zone typical for gluteus minimus referred pain during the SP test. The GS patients presented less intense but longer-lasting amplified vasodilatation in the pain region compared to SP-reactive sciatica subjects. It is not clear whether the SP-reactive sciatica patients were overlooked during the pressure palpation that was performed to confirm an overlapping GS diagnosis or if those patients presented with other subtypes of nociplastic pain.

## Figures and Tables

**Figure 1 jcm-10-05146-f001:**
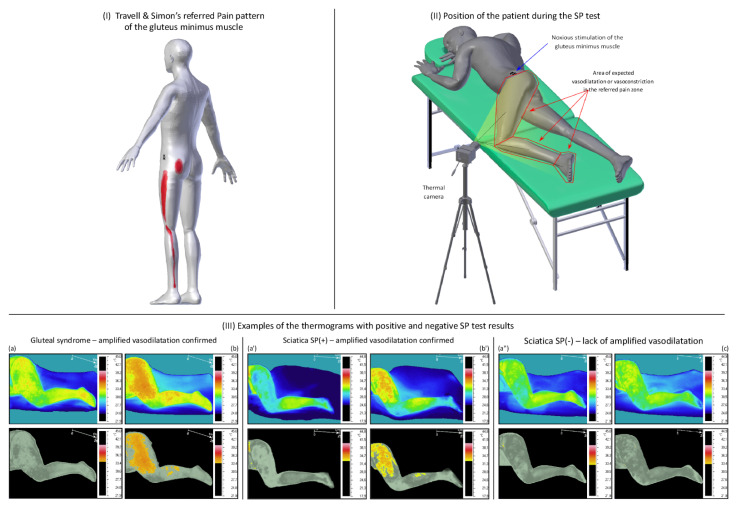
An illustration of the SP test applied to a gluteal syndrome and chronic sciatica. Legend: (**I**) gluteus minimus-referred pain pattern established by Travell and Simons based on the post-stimulation reports of the patient with gluteus minimus trigger points; (**II**) position of the patient during the SP test; and (**III**) examples of the thermograms with positive and negative SP test results; (**a**,**a’**,**a”**) pictures of a patient without noxious stimulation; (**b**,**b’**) pictures showing positive SP test results—registered amplified vasodilatation within the patient’s perceived pain zone; and (**c**) pictures showing negative SP test results—lack of amplified vasodilatation within the patient’s perceived pain zone (color picture—full range of temperature; gray picture—individually isolated state above the maximum baseline temperature at rest).

**Figure 2 jcm-10-05146-f002:**
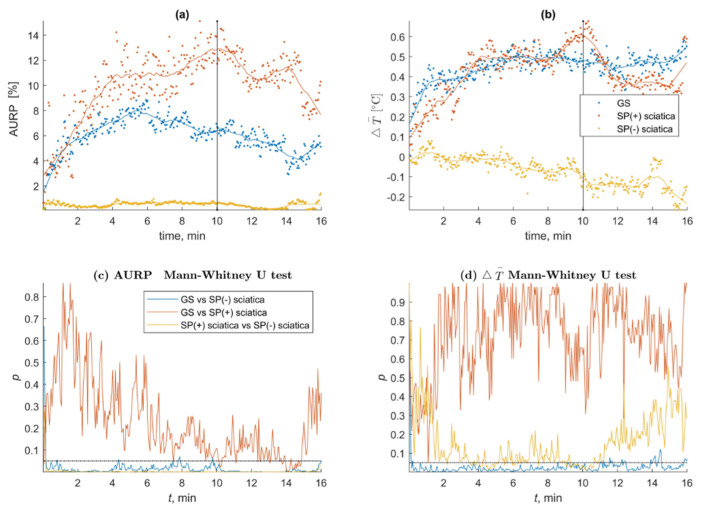
MATLAB trends and results: significance every 3 s for Δ₸° and AURP_T0_ of the GS and sciatica patients depending on SP test sensitivity; (**a**) MATLAB trends for the AURP results of the GS and sciatica patients; (**b**) MATLAB trends for the average temperature changes of the GS and sciatica patients; (**c**) MATALB trends of the statistical significance for AURP tested every out of Δ₸° 320 thermograms of the GS patients vs. sciatica and between sciatica patients depend on the SP test results; (**d**) MATALB trends of the statistical significance for Δ₸° tested every out of 320 thermograms of the GS patients vs. sciatica and between sciatica patients depend on the SP test results.

**Table 1 jcm-10-05146-t001:** Baseline characteristics of the participants (*n* = 50).

Data	Gluteal Syndrome (*n* = 20)	Sciatica (*n* = 30)	Sig (*p* = 95%)
Age, mean (SD) (y)	45.6 (± 6.0)	45.9 (± 7.6)	* 0.895
Pain intensity (VAS) (mm)	6.1 (± 2.4)	5.7 (± 1.5)	# 0.992
Symptoms duration mean (SD)(y)	9.6 (± 7.7)	14.5 (± 32.2)	* 0.508
1Q/Median/3Q	3.5/8.0/12.0	3.8/6.0/12.0	
Leg pain above knee (%)	*n* = 20 (100.0%)	*n* = 30 (100.0%)	** 1.000
Leg pain below knee (%)	*n* = 9 (45.0%)	*n* = 10 (33.3%)	** 0.553
Leg pain below ankle (%)	*n* = 3 (15.0%)	*n* = 5 (16.7%)	** 1.000
Gluteus minimus TrPs	20	0	-
Gluteus medius TrPs	5	0	-
Gluteus maximus TrPs	2	0	-
Quadratus lumborum TrPs	18	0	-

VAS: visual analog scale; TrPs: trigger points; SD: standard deviation; * *t*-test for independent groups; ** Fisher Exact test: # Independent-Samples Mann–Whitney significance.

**Table 2 jcm-10-05146-t002:** The main characteristics of the SP test results for the examined subgroups.

Group of the Patients with/without Muscle-Referred Pain (SP (+/−)) (*n* = …)	Characteristics of the Thermograms SP(+) with Confirmed Muscle-Referred Pain		
	First Thermogram at (min/s)	Last Thermogram at (min/s)	Number of Thermograms *n* = … (%)
GS (*n* = 20)	00′9″	16′00″	290 */** (90.6)
SP(+) Sciatica (*n* = 8)	2′12″	16′00″	66 */** (20.6)
SP(−) Sciatica (*n* = 22)	absence	absence	0 (0.0)

GS—gluteal syndrome; * *p* value < 0.005; SP—Skorupska protocol; Independent-Samples Mann–Whitney U Test; ** *p*-value < 0.05; Dunn Test (with Bonferroni adjustment for multiple comparisons).

**Table 3 jcm-10-05146-t003:** The summary of the SP test parameters according to the positive-SP-test definition depending on the test phase and examined subgroup.

SP	Phases of the SP Test	Group	Average (SD)	1Q/Median/3Q	Out of 320 SP(+) *n* = (%)	Significance of Difference between Groups
Δ₸°	the stimulation phase (10′)	GS	0.5 (0.6)	0.0/0.3/1.0	GS vs. SP(−) Sciatica 195 (97.5) GS vs. SP(+) Sciatica 1 (0.5) SP(+) Sciatica vs. SP(−) Sciatica 52 (26.0)	GS vs. SP(−) Sciatica 0.031 * GS vs. SP(+) Sciatica 0.285 SP(+) Sciatica vs. SP(−) Sciatica 0.039 *
		SP(−) Sciatica	−0.1 (0.7)	−0.7/−0.1/0.4		
		SP(+) Sciatica	0.6 (0.7)	0.3/0.5/0.9		
	the observation phase (16′)	GS	0.6 (0.7)	0.3/0.7/1.0	GS vs. SP(−) Sciatica 95 (79.2) GS vs. SP(+) Sciatica 0 (0.0) SP(+) Sciatica vs. SP(−) Sciatica 14 (11.7)	GS vs. SP(−) Sciatica 0.026 * GS vs. SP(+) Sciatica 0.500 SP(+) Sciatica vs. SP(−) Sciatica 0.216
		SP(−) Sciatica	−0.2 (0.9)	−0.7/−0.2/0.5		
		SP(+) Sciatica	0.6 (0.8)	0.2/0.6/1.0		
AURP_T0_	the stimulation phase (10′)	GS	6.7 (11.4)	0.3/2.9/7.4	GS vs. SP(−) Sciatica 192 (96.0) GS vs. SP(+) Sciatica 2 (1.0) SP(+) Sciatica vs. SP(−) Sciatica	GS vs. SP(−) Sciatica 0.019 * GS vs. SP(+) Sciatica 0.040 * SP(+) Sciatica vs. SP(−) Sciatica 0.001*
		SP(−) Sciatica	0.6 (1.1)	0.1/0.3/1.2		
		SP(+) Sciatica	11.7 (10.5)	1.9/10.8/22.0		
	the observation phase (16′)	GS	6.6 (8.7)	0.6/4.2/12.2	GS vs. SP(−) Sciatica 120 (100.0) GS vs. SP(+) Sciatica 27 (22.5) SP(+) Sciatica vs. SP(−) Sciatica 120 (100.0)	GS vs. SP(−) Sciatica 0.047 * GS vs. SP(+) Sciatica 0.158 SP(+) Sciatica vs. SP(−) Sciatica 0.034 *
		SP(−) Sciatica	1.5 (4.3)	0.0/0.1/0.4		
		SP(+) Sciatica	10.4 (8.9)	6.5/12.2/15.3		

GS—gluteal syndrome; SP—Skorupska protocol; AURP_T0_—autonomic referred pain; Δ₸—delta of the average temperature increase; SP(−)—negative test; SP (+)—positive test; * *p*-value < 0.05 Dunn post hoc test with Holm–Sidak correction; 1Q—first quartile; 3Q—third quartile.

## Data Availability

The data are not publicly available due to data privacy regulations. The data presented in this study are available upon request from the corresponding author.

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
