# Peer review of "Amplified Vasodilatation within the Referred Pain Zone of Trigger Points Is Characteristic of Gluteal Syndrome—A Type of Nociplastic Pain Mimicking Sciatica"

_jcm, 2021, doi:10.3390/jcm10215146_

Round 1
Reviewer 1 Report
There have been a sensible improvement in the manuscript in such a way I am now able to have an idea of the design and the sense of the results. However there are still a lot of concerns, some about the things are presented, and some about how the data are treated. I hope that you will accept this effort I request, but I really think it is mandatory.
Firstly, wording and definitions. I do insist saying that “Minimally Invasive Procedure” is a too general term for a very special and new functional exploration. In your response you call it “active dynamic thermography”, why not doing it in the paper? (or whatever else you prefer, “remote/referred thermography” or “remote/referred vasodilatation test”…). But this is not the main issue. You finally compare two groups of patients (GS and CS), and how you categorised them seems fine (but I am no rheumatologist or neurologist). You defined that two parameters from the test are used to define a positive response, ΔTavr and AURP: fine, but how do you define a positive response? Firstly, it seems obvious that a response (positive or not) applies to an observation (i.e. one active thermogram into one ROI), not to a patient. I agree that a patient could be further defined as a responder to the test (or not: what you named MIP+/–) from the results of several observations in several ROIs, but you have to explain how. Furthermore, I am not able to see how and observation is defined as positive: which level of increase of Tavr or size of AURP are needed to be “significant” (although this word hardly applies to a single observation, see below)? To define a positive response, do you need to have increase in Tavr, AURP or both? If you are not able to provide such definitions of a positive response, you are constrained to present only raw values of changes, and to compare them between groups – although I do not know how you treat the issue of ROI: by merging all the ROIs, by comparing ROI by ROI?
Secondly, the manuscript is still too long. The primary objective seems to be to differentiate GS from CS by this new exploration, while clinical examination was apparently insufficient to do so. So it is not the validation of the new exploration: then there is no need to extend so much the presentation of the technique and of the signal treatment, especially as you apparently already did so in another paper. The manuscript must focus on the primary objective, not ramble around. In line with this, as you only observed GS and CS patients, there are no healthy controls, and it comforts me in the idea that it is the development of a diagnosis tool. The absence of healthy controls is actually a limitation in the study, not critical if we keep a pragmatic approach (the physician would explore only symptomatic people), but to be discussed. Did you conduct case/control studies in your previous researches?
All my following point-by-point comments illustrate these two major concerns I have: data treatment and presentation.
- The title is a bit too general; “amplified autonomic phenomenon” is actually vasodilatation, and the “new chronic low back pain subtype mimicking sciatica” is actually gluteal syndrome. “A retrospective analysis supported by MATLAB” is off-topic and useless.
- The abstract has to be totally rewritten. It does not summarise well neither the methods, nor the results, see below;
- Introduction, explain the acronym TrPs at its first mention.
- Introduction, page 3, you wrote “(i) the size of the autonomic referred pain (AURP) (temporarily amplified vasomotor reactivity zone) associated with (ii) significant average temperature (ΔTavr) changes in that zone.” So you will have to define in the Methods what you call “significant” (see above); it seems to be an increase in T°, not a change (a change could be a decrease, i.e. vasoconstriction); I personally prefer ΔavgT° or Δ₸° to Δ
- Just after, you wrote “The MIP results were considered positive (MIP(+)) if a simultaneous increase in both parameters was confirmed, with ΔTavr as the leading indicator of amplified vasomotor activity.” So you need to have both an increased T° and an AURP (or an increased AURP?) to define a response? I guess this applies to an observation, not to a patient (see above)? If T° is a “leading indicator”, does it mean it weighs more than the AURP? If so, how?
- Just after, you wrote “Additionally, according to our previous studies, the autonomic phenomenon was confirmed if a specific muscle had active trigger points and the patients reported increased pain sensation during the MIP.” Was this confirmation necessary to define the response? It seems there is a confusion with the definition of triggered pain, see below.
- The end of the introduction about the needle and MATLAB are off-topic; anyway, an introduction should not exceed 500 words.
- In Methods page 4 you wrote “A retrospective analysis of the thermal data extracted from previously published studies”: it is not a retrospective analysis, it is an ancillary study or a secondary analysis.
- End of 2.1 you wrote “The sciatica patients who reported pain provocation in the symptomatic region during the stimulation phase of the procedure were classified as having MIP-sensitive (MIP(+)) sciatica. The sciatica patients not reactive during the gluteus minimus noxious stimulation were classified as not sensitive to MIP (MIP(-)).” So I must understand that all GS patients were naturally what you call “MIP(+)”; if this is true, it has to be stated. Furthermore, the term “MIP(+/–)” is very confusing, firstly because it contradicts your previous definition of “MIP(+/–)”, and because it has nothing to do with the functional exploration per se. It is more a presence/absence of referred pain.
- The first sentence of 2.2 is useless as the information is already in the table.
- Inclusion criteria (2.2): first define the common ones for all patients (age, pain duration), then those specific to each group. For the straight-leg raise test, I guess you talk about degrees (>30°, <60°…)?
- To present the numerical data, one digit after dot is enough (except for P values, 3 digits).
- Table 1: VAS (rather “pain intensity (VAS)”) an duration of symptoms must be compared with Mann-Whitney’s tests.
- Table 1: I did not know Z-test was to compare proportions… Usually it is the chi-square, or rather in your case, the Fisher’s exact test…
- I do not think the flow chart is useful, this is neither a clinical trial nor a cohort study.
- Chapter 2.3.1 is too long (see above), especially the MATLAB parts (MATLAB is just one way to treat signals); a part could be transferred to appendixes. On the other hand, the definition of positive observations is not given…
- Chapter 2.5 you wrote “Tests were applied to compare the differences in AURP and ΔTavr between the afore-mentioned groups.” Tests do not compare differences, they either test differences of they compare data.
- Chapter 2.5 you wrote “Values, figures, and tables in the text are expressed as the means ± standard deviation (SD).” This is acceptable only for Gaussian-distributed variables. I doubt they were, since you used Mann-Whitney’s tests… So show the quartiles for such variables.
- The way you present your results is still very confusing and illustrate my general impression. Firstly, it seems that the final comparison between the two groups lies on counting the number of positive responses (thermograms with “significant” triggered vasodilation), so was this the primary outcome? Such approach could be acceptable, but in this case you must adjust to the number of patients and the number of observations per patient, if these are not balanced. Interestingly, it seems that many more observations (patients?) were positive in the GS group, which is totally unsaid in the abstract, and which furthermore is the main result (if the aim was to develop a diagnosis tool to separate GS from CS).
- The results are even less understandable when you talk about the “MIP(–) sciatica” cases: how many were they out of the 30? Are they still counted in the whole CS of your abstract?
- My opinion is unchanged about your table 2 (see previous review): change it to a curve.
- The 3.1.1 to 3 chapters are interesting but take too much place; results would be more readable in a Table. The concern about the two subgroups of CS is unchanged (how many cases in each?), and probably there should be in addition a description of the whole CS group.
- In those chapters is the first time you mention a stimulation and an observation phase, this has to be explained in the Methods, and how these two phases have to be interpreted.
- In those chapters, you define as subtitles “the 1st / 2nd MIP parameters”: please name them by their name! (ΔavgT° or Δ₸°, AURP).
- Is the information about the greatest effects/differences at some time point very useful?
Author Response
Dear Reviewer 1,
We would like to thank you greatly for the time and effort you put into reviewing our paper. All of your suggestions forced us to work hard to improve the paper, especially with respect to the protocol definition and results presentation. Your doubts and questions helped us define the negative/positive test results more precisely. Also, the results are now presented more clearly and accurately. Thank you a lot for the time and patience you took to get familiar with our study.
Please find our response to your suggestions below:
Firstly, wording and definitions. I do insist saying that “Minimally Invasive Procedure” is a too general term for a very special and new functional exploration. In your response you call it “active dynamic thermography”, why not doing it in the paper? (or whatever else you prefer, “remote/referred thermography” or “remote/referred vasodilatation test”…). But this is not the main issue.
We agree with your opinion. We have replaced the Minimally Invasive Procedure with Skorupska Protocol (SP).
We were considering the name nociceptive Active Dynamic Thermography. However, due to the fact that SP includes lots of innovative solutions, our international expert in the field of medical thermography suggested that we use Skorupska Protocol, which he believed to be a more suitable term.
You finally compare two groups of patients (GS and CS), and how you categorized them seems fine (but I am no rheumatologist or neurologist). You defined that two parameters from the test are used to define a positive response, ΔTavr and AURP: fine, but how do you define a positive response?
Firstly, it seems obvious that a response (positive or not) applies to an observation (i.e. one active thermogram into one ROI), not to a patient. I agree that a patient could be further defined as a responder to the test (or not: what you named MIP+/–) from the results of several observations in several ROIs, but you have to explain how. Furthermore, I am not able to see how and observation is defined as positive: which level of increase of Tavr or size of AURP are needed to be “significant” (although this word hardly applies to a single observation, see below)? To define a positive response, do you need to have increase in Tavr, AURP or both? If you are not able to provide such definitions of a positive response, you are constrained to present only raw values of changes, and to compare them between groups – although I do not know how you treat the issue of ROI: by merging all the ROIs, by comparing ROI by ROI?
Our response:
A positive test is confirmed if within the observed ROI with possible referred pain occurrence a simultaneous change in both parameters is observed, i.e., (i) the development of AURPTmax and (ii) an increase in (Δ₸°) of more than 0.3°C at the end of the stimulation phase and during the observation phase.
The definition is based on:
- A standard in medical thermography for musculoskeletal system dysfunctions which recognizes thermal asymmetries greater than 0.3–0.7 °C.
- The validation study of the test (TrPs positive sciatica vs TrPs negative sciatica vs healthy control)
Validity and reliability for thigh ROIs
|
|
phase |
sensitivity |
specificity |
|
|
Δ₸° |
stimulation |
100 |
100 |
|
|
observation |
100 |
84.6 |
|
|
|
AURPTmax |
stimulation |
100 |
100 |
|
|
observation |
100 |
100 |
|
Secondly, the manuscript is still too long. The primary objective seems to be to differentiate GS from CS by this new exploration, while clinical examination was apparently insufficient to do so. So it is not the validation of the new exploration: then there is no need to extend so much the presentation of the technique and of the signal treatment, especially as you apparently already did so in another paper. The manuscript must focus on the primary objective, not ramble around. In line with this, as you only observed GS and CS patients, there are no healthy controls, and it comforts me in the idea that it is the development of a diagnosis tool. The absence of healthy controls is actually a limitation in the study, not critical if we keep a pragmatic approach (the physician would explore only symptomatic people), but to be discussed. Did you conduct case/control studies in your previous researches?
You are right. The part about Matlab has been removed (supplementary matierials). The lack of the control group for the present study has been added as a limitation of the study.
All my following point-by-point comments illustrate these two major concerns I have: data treatment and presentation.
- The title is a bit too general; “amplified autonomic phenomenon” is actually vasodilatation, and the “new chronic low back pain subtype mimicking sciatica” is actually gluteal syndrome. “A retrospective analysis supported by MATLAB” is off-topic and useless.
A new proposal of the title: Amplified vasodilatation within the referred pain zone of trigger points is characteristic of gluteal syndrome – a type of nociplastic pain mimicking sciatica
We have decided to be more precise and would like to explain what gluteal syndrome (GS) is already in the title to give a hint at what we examined. In the older literature, the term deep gluteal syndrome was used, e.g., to describe a type of sciatic nerve entrapment. Thus, a possible misunderstanding could arise.
- The abstract has to be totally rewritten. It does not summarise well neither the methods, nor the results, see below;
The abstract has been rewritten.
- Introduction, explain the acronym TrPs at its first mention.
It has been corrected.
- Introduction, page 3, you wrote “(i) the size of the autonomic referred pain (AURP) (temporarily amplified vasomotor reactivity zone) associated with (ii) significant average temperature (ΔTavr) changes in that zone.” So you will have to define in the Methods what you call “significant” (see above); it seems to be an increase in T°, not a change (a change could be a decrease, i.e. vasoconstriction); I personally prefer ΔavgT° or Δ₸° to Δ
The definition of a positive test has been added.
We have changed ΔTavr to Δ₸°.
- Just after, you wrote “The MIP results were considered positive (MIP(+)) if a simultaneous increase in both parameters was confirmed, with ΔTavr as the leading indicator of amplified vasomotor activity.” So you need to have both an increased T° and an AURP (or an increased AURP?) to define a response? I guess this applies to an observation, not to a patient (see above)? If T° is a “leading indicator”, does it mean it weighs more than the AURP? If so, how?
Thank you very much for these questions. They showed us the gaps in our method description.
The definition of a positive test was provided above. Following your question, we have added this description to the method:
Single case test results: The main test result is the occurrence of a new isotherm (AURPTmax) in the area of expected referred pain. The average temperature increase always follows AURP.
Group examination: The gluteus minimus muscle can refer pain to the thigh, calf, and rarely to the foot. Depending on a case, patients can present either full or partial pain patterns. Δ₸° is the leading SP test factor to determine the number of subjects with muscle-referred pain at a specific ROI. The previous study revealed that if TrPs presented calf referred pain for around 40% of the group, the method showed negative test results for that ROI because only AURP was significant, while Δ₸° was not.
Test application to compare the examined groups of subjects
To calculate the significance of the diagnostic parameters in a group of subjects, the thermograms of the examined subgroup were compared every 3 seconds of the test and then the significant differences for every parameter were calculated. Next, the time of the test when both AURP and Δ₸° were confirmed significant was marked as referred pain significant for the specific ROI.
- Just after, you wrote “Additionally, according to our previous studies, the autonomic phenomenon was confirmed if a specific muscle had active trigger points and the patients reported increased pain sensation during the MIP.” Was this confirmation necessary to define the response? It seems there is a confusion with the definition of triggered pain, see below.
The patient report is additional information that allowed us to state that the method reflects pain provoked at the moment of the stimulation. Some of the cases, who felt daily pain on the thigh and calf, reported pain sensation only on the thigh during the test. Then, the thermograms showed AURP only on the thigh. This confirms that during the test we measured the parameters at the exact moment of inducing referred pain by needling.
- The end of the introduction about the needle and MATLAB are off-topic; anyway, an introduction should not exceed 500 words.
That part has been removed.
- In Methods page 4 you wrote “A retrospective analysis of the thermal data extracted from previously published studies”: it is not a retrospective analysis, it is an ancillary study or a secondary analysis.
We have corrected the mistake.
- End of 2.1 you wrote “The sciatica patients who reported pain provocation in the symptomatic region during the stimulation phase of the procedure were classified as having MIP-sensitive (MIP(+)) sciatica. The sciatica patients not reactive during the gluteus minimus noxious stimulation were classified as not sensitive to MIP (MIP(-)).” So I must understand that all GS patients were naturally what you call “MIP(+)”; if this is true, it has to be stated. Furthermore, the term “MIP(+/–)” is very confusing, firstly because it contradicts your previous definition of “MIP(+/–)”, and because it has nothing to do with the functional exploration per se. It is more a presence/absence of referred pain.
We have added the following sentences to the paper: Line 108 “The gluteal syndrome patients presented active gluteus minimus trigger points and reported pain sensation during the test. Thus, they were classified as SP(+).”
In the method section (2.3.1), we have added the following sentence: The presence of muscle-referred pain is marked as SP(+) and the lack of muscle-referred pain is marked as SP(-). (line 166)
- The first sentence of 2.2 is useless as the information is already in the table.
The first sentence has been removed.
- Inclusion criteria (2.2): first define the common ones for all patients (age, pain duration), then those specific to each group. For the straight-leg raise test, I guess you talk about degrees (>30°, <60°…)? To present the numerical data, one digit after dot is enough (except for P values, 3 digits). Table 1: VAS (rather “pain intensity (VAS)”) an duration of symptoms must be compared with Mann-Whitney’s tests.
All the indicated corrections have been made.
- Table 1: I did not know Z-test was to compare proportions… Usually it is the chi-square, or rather in your case, the Fisher’s exact test…
The mistake has been corrected.
- I do not think the flow chart is useful, this is neither a clinical trial nor a cohort study.
The flow chart has been removed.
- Chapter 2.3.1 is too long (see above), especially the MATLAB parts (MATLAB is just one way to treat signals); a part could be transferred to appendixes. On the other hand, the definition of positive observations is not given…
The description of the Matlab procedure has been moved to an appendix.
- Chapter 2.5 you wrote “Tests were applied to compare the differences in AURP and ΔTavr between the afore-mentioned groups.” Tests do not compare differences, they either test differences of they compare data.
The sentences have been changed according to your suggestion.
Chapter 2.5 you wrote “Values, figures, and tables in the text are expressed as the means ± standard deviation (SD).” This is acceptable only for Gaussian-distributed variables. I doubt they were, since you used Mann-Whitney’s tests… So show the quartiles for such variables.
We are very grateful for your notice. It is true that the variables do not meet normality distribution assumption and it is why we used the Mann-Whitney test. We have corrected the table 3 and presented results. However, we have decided that according to the literature we would like to report also the mean and standard deviation because of their usefulness during summarizing data with other researchers or future meta analyses.
- The way you present your results is still very confusing and illustrate my general impression. Firstly, it seems that the final comparison between the two groups lies on counting the number of positive responses (thermograms with “significant” triggered vasodilation), so was this the primary outcome? Such approach could be acceptable, but in this case you must adjust to the number of patients and the number of observations per patient, if these are not balanced. Interestingly, it seems that many more observations (patients?) were positive in the GS group, which is totally unsaid in the abstract, and which furthermore is the main result (if the aim was to develop a diagnosis tool to separate GS from CS).
The definition of a positive test you insisted on adding helped us to sort and present the data more clearly.
The results presentation has been corrected.
- The results are even less understandable when you talk about the “MIP(–) sciatica” cases: how many were they out of the 30? Are they still counted in the whole CS of your abstract?
The number of sciatica cases with positive and negative tests has been added.
- My opinion is unchanged about your table 2 (see previous review): change it to a curve.
It has been changed according to your suggestion.
In the end, we have decided to remove that part. The data measured every 3 seconds are provided in supplementary materials. Thanks to your suggestion of defining a positive test, it is now more suitable to present AURP and Δ₸° data as we did in the new Table 3, where we presented the summary of positive SP test results according to the definition.
- The 3.1.1 to 3 chapters are interesting but take too much place; results would be more readable in a Table. The concern about the two subgroups of CS is unchanged (how many cases in each?), and probably there should be in addition a description of the whole CS group.
The results section has been rewritten and more of the data has been presented in the form of a table.
- In those chapters is the first time you mention a stimulation and an observation phase, this has to be explained in the Methods, and how these two phases have to be interpreted.
We have added necessary information in chapter 2.3.1, as well as a new definition of positive test results.
- In those chapters, you define as subtitles “the 1st / 2nd MIP parameters”: please name them by their name! (ΔavgT° or Δ₸°, AURP).
We have removed the confusing notation (1st/2nd MIP parameter) and introduced AURP and Δ₸° to the description.
- Is the information about the greatest effects/differences at some time point very useful?
You are right. The data are useless for the audience at the moment. We need more studies to answer if the AURP size has clinical meaning. We would like to show numerically that the observed phenomenon has strobing nature before it develops a more stable character. We have attached a video file to present that. The video reflects our observation during the test: if the needle provoked a sensitized nociceptor – patients reported referred pain, muscle presented twitch response, and the infrared camera registered vasomotor reactivity. At the beginning, the reaction was very brisk and short-lasting. If the next twitch was evoked soon enough, the phenomenon occurred as a series. However, some of the cases presented the weak form of TrPs – then, the reactions registered by the camera were weak as well.
We speculate that the amount of the thermograms with confirmed muscle-referred pain can have clinical meaning in the future. Thus, these data are provided in new tables.

Reviewer 2 Report
Dear authors,
Thanks you for the corrections and the new argument in the introduction. The answers are satisfactory and I do not have more comments.
Kind regards
Author Response
Dear Reviewer,
Thank you very much for the time and effort you spent reviewing our paper. We greatly appreciate your kind opinion.
Kind regards,
authors
Round 2
Reviewer 1 Report
Despite the improvements made (mostly in manuscript size and clarity in some parts), I am still very far from understanding well the study design. Your changes in the abstract, the introduction and the methods parts 2.1 and 2.2, as well in Tables 1-2, are appreciated. For several points however, my previous comments are still valuable. I must warn you that if no sensible effort is done in presentation I shall not go further in my reviewing.
My major concern is still in the description of the exploration and the definitions of positivity. Let us take the lines 107-110 and your chapter 2.3.1 (*/* signals where my comment starts):
- “The gluteal syndrome patients presented active gluteus minimus trigger points and reported pain sensation during the test. Thus, they were classified as SP(+).” */* So I understand that evoked pain during testing is the main outcome to define positivity; so given the ICD definition of GS, they should all be SP(+)…
- “The sciatica patients who reported pain provocation in the symptomatic region during the stimulation phase of the procedure were classified as having SP-sensitive (SP(+)) sciatica.” */* So this is still defined by evoked pain, but only during the “procedure” (the exploration?); why is it different for them?
- “A positive SP test result is based on a temporary autonomic nervous system imbalance resulting in amplified vasomotor reactivity within the referred pain zone of active trigger points.” */* This contradicts what you wrote in chapter 2.1; now you need vasodilation…
- “The test consists of two parts: (i) noxious, nociceptive muscle stimulation under thermal camera control of the area with expected referred muscle pain (10 minutes);” */* OK, but describe the stimulation; was it continuous, intermittent? How did you calibrate the pressure on muscles? How did you localize the muscles? Did all the subjects bore the 10min-session? What happened if they asked for stopping? By “expected” did you mean the same region as for the screening?
- “(ii) further observation of patients at rest using a thermal camera (6 minutes).” */* Rather name this “post-stimulation resting phase” (not “observation” as you were observing before).
- “Two diagnostic parameters were defined: (i) the occurrence of a new isotherm named autonomic referred pain (AURP) reflecting the percentage size of induced vaso-motor reactivity within the observed region of interest (ROI)…” */* So here – at last! – I understand that “AURP” is actually a thermographic parameter. OK, but (i) what means “new”: something that was not observed at baseline?; (ii) “isotherm” means to me that T° does not change; (iii) AURP is not a good wording as there is no pain in it; and (iv) a percentage must be defined by its denominator. You did not define the ROI either (I guess it must be a painful area…).
- “… and (ii) changes in the average temperature, where the difference between the average temperature at rest and the average temperature measured every 3 seconds of the procedure was defined as the delta of the average temperature (Δ₸°).” */* So Δ₸° is the other thermographic parameter; I guess there is one measurement every 3s, and that “at rest” was a baseline value before stimulation, am I right? It seems that Δ₸° and “AURP” are just the same parameter ₸° treated differently from a baseline (similar or different)?
- “The AURP isotherm had two components: (i) AURP above the maximum temperature (Tmax) named AURPTmax, and (ii) AURP below Tmax.” */* What is Tmax? Maximal of what? How do you treat the “above” and the “below”, by an area-under-curve? Is the part “below” useful?
- “The validation study confirmed 100% sensitivity and 100% specificity for AURPTmax.“ */* Against which standard?
- “A positive test is confirmed if within the observed ROI with possible referred pain occurrence a simultaneous change in both parameters is observed, i.e., (i) the development of AURPTmax…” */* Is it the “SP” which is positive? What previous result do you “confirm”? The wording ‘if… occurrence a simultaneous change” is inappropriate. What means “development of AURPTmax”, development from what? If you need effects on both “AURP” and Δ₸°, I suppose there can be “sub-positive” cases with only one positive, how do you deal with it?
- “and (ii) an increase in Δ₸°of more than 0.3°C at the end of the stimulation phase and during the observation phase.” */* define the time frame of “at the end” and of “during”, and the necessary duration of the event to be positive.
- NB if there is an event “at the end” but not during the post-stimulation resting phase, how do you deal with it?
- “The presence of muscle-referred pain is marked as SP(+) and the lack of muscle-referred pain is marked as SP(-).” */* So this corresponds to the 1st definition, with the same lack of precision.
- “Single case test results: The main test result is the occurrence of a new isotherm (AURPTmax) in the area of expected referred pain.” */* And this is totally different (positivity defined by thermography).
- “The average temperature increase always follows AURP.” */* So if it always increase, is there a need for two parameters?
- “Group examination: The gluteus minimus muscle can refer pain to the thigh, calf, and rarely to the foot.” */* OK, so you can have up to 3 ROIs; is one enough to define a positive subject?
- “Depending on a case, patients can present either full or partial pain patterns. Δ₸° is the leading SP test factor to determine the number of subjects with muscle-referred pain at a specific ROI.” */* ??? so a single observation would determine a number? So temperature would define pain?
- “To calculate the significance of the diagnostic parameters in a group of subjects, the thermograms of the examined subgroup were compared every 3 seconds of the test and then the significant differences for every parameter were calculated.” */* ??? I just cannot understand this. NB you do not calculate a significance, you test it. What is a group, a subgroup?
- “Next, the time of the test when both AURP and Δ₸° were confirmed significant was marked as referred pain significant for the specific ROI.” */* “Referred pain significant” has no meaning to me. Neither “the time to the test”. I guess it what the time of occurrence of the first event during the stimulation phase, but I am not sure.
- “A simultaneous significant increase or decrease in the two parameters determining the area with amplified vasodilatation or vasoconstriction within trigger-point referred pain provoked during the SP is coincident with the patient’s daily complaint.” */* So there can be decrease/vasoconstriction too? This is new!
Other comments:
- This is a test/phenotype relationship study to validate a diagnosis tool.
- Table 1 (which is a result, not a method): I doubt that the duration of symptoms follow a Gaussian distribution.
- “2.3 Method”: §1 is actually “Confirmation of phenotypes”.
- “2.3.1 Skorupska Protocol” is actually “2.4 Functional exploration”.
- “Skorupska Protocol is a new validated test proposed to objectively confirm nociplastic pain related to muscle.” */* (1) Name it so if you like, but I prefer what I suggested (and probably the journal too); by doing so, Mrs Skorupska will not be able to name so any other test or exploration she develops. (2) What do you mean by “validated”? It seems that here is the validation study. (3) This does not refer to my definition of nociplastic; it looks more like sensitisation, which is nociceptive; fibromyalgia is nociplastic.
- Statistical methods: there is no Z value (or I missed something).
- I cannot understand the legends of Fig.1. It is not “visualization”, it is an illustration.
- Reduce the discussion to 1500 words and only focus on
- Summary of the main results
- Comparison to similar studies if any
- Explanations for the results (pathophysiology of GS)
- Biases and limitations
- Consequences for current care or future research
- No results in the discussion.
- NB I will read thoroughly the other tables and the discussion when the rest is addressed.
Author Response
Dear Reviewer 1,
Having read your review, we reanalyzed all the comments and suggestions you have made so far. We think that we made a mistake by providing a purely technical description of AURP and using the terminology known only to those specializing in myofascial pain or medical thermography. To clarify our description, in this new version of our paper we have explained that AURP reflects the percentage size of the observed phenomenon. It is an innovative approach to thermal data analysis, which has been limited so far to the analysis of temperature parameters, e.g., minimum, average, or maximum, and has not accounted for the size of induced reactions. However, the latter is clinically important because vasodilatation occurs exactly in the area of the reported pain dependent on the CS subtype. Up to now, there is no method that could enable the objective diagnosis of nociplastic pain associated with the presence of trigger points.
Following your suggestions, we have made an attempt at correcting all the inaccuracies in accordance with the suggestions you gave in the last review.
My major concern is still in the description of the exploration and the definitions of positivity. Let us take the lines 107-110 and your chapter 2.3.1 (*/* signals where my comment starts):
- “The gluteal syndrome patients presented active gluteus minimus trigger points and reported pain sensation during the test. Thus, they were classified as SP(+).” */*
So I understand that evoked pain during testing is the main outcome to define positivity; so given the ICD definition of GS, they should all be SP(+)
- “The sciatica patients who reported pain provocation in the symptomatic region during the stimulation phase of the procedure were classified as having SP-sensitive (SP(+)) sciatica.” */* So this is still defined by evoked pain, but only during the “procedure” (the exploration?); why is it different for them?
Answers 1 and 2
The SP test can confirm trigger points in any muscle. GS is one of the possibilities. All of the subjects were SP-positive. Following your doubts, we have tried to be more precise and in the revised version we have added what follows:
A positive test was confirmed if a patient with trigger points presented test-provoked vasodilatation and/or vasoconstriction in the daily complaint area coincident with the region of the referred pain pattern of the tested muscle. The autonomic phenomenon is defined by two parameters. The first parameter is a percentage size of the vasomotor reactivity subarea with the temperature not registered for the patient before the stimulation (T0), which was named autonomic referred pain (AURPT0). The second one is an increase in the delta of the average temperature (Δ₸°) in the examined body part of more than 0.3°C. Both diagnostic parameters were demanded to classify the test results as positive (SP(+)).
- “A positive SP test result is based on a temporary autonomic nervous system imbalance resulting in amplified vasomotor reactivity within the referred pain zone of active trigger points.” */*
This contradicts what you wrote in chapter 2.1; now you need vasodilation…
The SP test can provoke vasodilatation and/or vasoconstriction in the muscle-referred pain zone (SP(+)) depending on the tested muscle and the group of patients.
We can send you all the relevant data if you would like to get familiar with them.
- “The test consists of two parts: (i) noxious, nociceptive muscle stimulation under thermal camera control of the area with expected referred muscle pain (10 minutes);” */*
OK, but describe the stimulation; was it continuous, intermittent?
We have corrected the description and added a step by step list of the main parts of the SP test :
Description of the exploration
Both full and truncated descriptions of the method were presented in our previous studies [18,19]. The description of the MATLAB procedure for the SP test, which allowed the analysis of the test results every 3 seconds, was provided in supplementary materials.
Short description of the exploration:
- Trigger points examination based on the palpatory diagnostic criteria according to Travell and Simons
- Thermal side-to-side comparison of the patients at rest to exclude a possible neuropathic pain component (confirmed if in the pain subarea there was a temperature decrease of more than 0.5°C compared to the opposite side)
- Examination towards trigger points’ autonomic response at the daily complaint area coincident with the referred pain zone for the examined muscle
- Numerical analysis of the thermograms
The examination consists of two parts: (i) noxious, nociceptive muscle stimulation under thermal camera control of the area with expected muscle-referred pain (10 minutes) and (ii) post-stimulation resting phase – further thermal observation of patients at rest (6 minutes). For part (i), the noxious stimulation of the two most sensitive trigger points (GS) or the most tender muscle areas (sciatica) within the gluteus minimus muscle was applied using the continuous fast-in-fast-out dry needling technique. During the whole noxious stimulation, the patients reported the localization of the pain sensation provoked by needling (thigh, calf, foot).
Numerical analysis of the thermograms: A comparative analysis of the thermograms (320) registered every 3 seconds and the thermogram recorded for the patient before the stimulation was performed using MATLAB. For every thermogram, the thigh subarea was segmented. Then, both the AURPT0 size and Δ₸° of the thigh region were calculated. The significance of both diagnostic parameters was checked for every out of the 320 thermograms.
- How did you calibrate the pressure on muscles? How did you localize the muscles? Did all the subjects bore the 10min-session? All the subject underwent the same procedure (exploration). What happened if they asked for stopping? By “expected” did you mean the same region as for the screening?
It was not necessary to calibrate the pressure on the muscle. For every study in the field, the diagnostic palpatory criteria according to Travell and Simson are used to confirm the presence of trigger points. The specific value of muscle pressure is not included in the diagnostic criteria.
We have added the following sentences to the paper:
All of the participants finished the diagnostic protocol.
The participants had the right to refuse the SP test and withdraw from the study at any time without penalty.
- “(ii) further observation of patients at rest using a thermal camera (6 minutes).” */*
Rather name this “post-stimulation resting phase” (not “observation” as you were observing before).
It has been corrected.
- “Two diagnostic parameters were defined: (i) the occurrence of a new isotherm named autonomic referred pain (AURP) reflecting the percentage size of induced vaso-motor reactivity within the observed region of interest (ROI)…” */* So here – at last! – I understand that “AURP” is actually a thermographic parameter. OK, but (i) what means “new”: something that was not observed at baseline?; (ii) “isotherm” means to me that T° does not change; (iii) AURP is not a good wording as there is no pain in it; and (iv) a percentage must be defined by its denominator. You did not define the ROI either (I guess it must be a painful area…).
We have added the following:
Detailed AURPT0 explanation:
Autonomic referred pain (AURPT0) reflects the size of the phenomenon revealing the temperature not registered for the patient before the stimulation. Technically, AURPT0 is an isolated region (segmented part of the thermogram) with a defined range of temperature measured in the ROI every 3 seconds of the procedure individually for every case. Originally, the AURPT0 size (cm2) was calculated manually by a technician and then the percentage size of the area within the thigh, calf, and foot was calculated separately. That method allowed the AURP size calculation at the end of both test phases. Currently, the new codes created for the purposes of the test in the MATLAB environment allowed an automated AURP size calculation for every out of the 320 recorded thermograms.
- “… and (ii) changes in the average temperature, where the difference between the average temperature at rest and the average temperature measured every 3 seconds of the procedure was defined as the delta of the average temperature (Δ₸°).” */* So Δ₸° is the other thermographic parameter; I guess there is one measurement every 3s, and that “at rest” was a baseline value before stimulation, am I right? It seems that Δ₸° and “AURP” are just the same parameter ₸° treated differently from a baseline (similar or different)?
Both measurements are different than the baseline, now defined as T0.
As said above – AURP – the size of the induced phenomenon;
Δ₸° – the level of temperature changes compared to the baseline (T0); additionally this parameter is sensitive to the number of subjects with pain in that specific region
For example, every gluteus minimus trigger point refers pain to the thigh and sometimes to the calf. We proved that vasodilatation induced in n=9/20 TrPs-positive cases was not enough to provoke a significant increase in Δ₸° in the calf ROI. However, the AUROPT0 results were proven to be significant
- “The AURP isotherm had two components: (i) AURP above the maximum temperature (Tmax) named AURPTmax, and (ii) AURP below Tmax.” */* What is Tmax? Maximal of what? How do you treat the “above” and the “below”, by an area-under-curve? Is the part “below” useful?
You are right, it was confusing. We have removed the validation study data and the full method description, which was unnecessary. The validation study confirmed 100% sensitivity and 100% specificity for AURP Tmax only. In the revised version, AURPTmax was renamed AURPT0.
- “The validation study confirmed 100% sensitivity and 100% specificity for AURPTmax.“ */* Against which standard?
AURP is a new parameter created for the purposes of the method (SP test). It was defined above in point 6.
According to the validation study (References no 18), the SP test provoked an autonomic phenomenon dependent on the trigger points presence exclusively. The healthy subjects and pain patients without TrPs revealed no SP-provoked vasomotor activity within the pain zone.
- “A positive test is confirmed if within the observed ROI with possible referred pain occurrence a simultaneous change in both parameters is observed, i.e., (i) the development of AURPTmax…” */* Is it the “SP” which is positive? What previous result do you “confirm”?
The wording ‘if… occurrence a simultaneous change” is inappropriate. What means “development of AURPTmax”, development from what? If you need effects on both “AURP” and Δ₸°, I suppose there can be “sub-positive” cases with only one positive, how do you deal with it?
It is not possible to develop AURP without a simultaneous temperature increase.
However, you are right that our previous explanation of Δ₸° was confusing.
We have added a more precise definition (provided above in answers no. 1 and 2).
- “and (ii) an increase in Δ₸°of more than 0.3°C at the end of the stimulation phase and during the observation phase.” */* define the time frame of “at the end” and of “during”, and the necessary duration of the event to be positive.
We just needed to confirm the autonomic phenomenon. The data obtained for the phases are important to medical thermography specialists only. Thus, the definition has been corrected.
- NB if there is an event “at the end” but not during the post-stimulation resting phase, how do you deal with it?
The definition has been simplified.
- “The presence of muscle-referred pain is marked as SP(+) and the lack of muscle-referred pain is marked as SP(-).” */* So this corresponds to the 1st definition, with the same lack of precision.
The text has been corrected.
- “Single case test results: The main test result is the occurrence of a new isotherm (AURPTmax) in the area of expected referred pain.” */*
And this is totally different (positivity defined by thermography).
The new definition has made this paragraph redundant.
- “The average temperature increase always follows AURP.” */* So if it always increase, is there a need for two parameters?
We explained above that AURPT0 is the size of the vasomotor activity and Δ₸° is the level of temperature increase – both compared to the baseline thermogram. Both are necessary and different.
Moreover, our solution to numerically calculate the size with the pathological reaction within the pain zone is a completely new parameter in medical thermography.
The manual thermogram segmentation was the authors’ solution. The same is true for automatic Matlab analysis, which demanded the new codes and programs written by an engineer in the Matlab environment (there is no MATLAB app dedicated to thermogram segmentation needed for the SP test). We are going to create dedicated software for thermogram segmentation in the future.
- “Group examination: The gluteus minimus muscle can refer pain to the thigh, calf, and rarely to the foot.” */* OK, so you can have up to 3 ROIs; is one enough to define a positive subject?
Yes,
We explained that shortly above.
Our last study showed that subjects with gluteus minimus TrPs refer pain to the thigh, the test results of the calf and foot were negative. It means that they had TrPs referred pain to the thigh only, but they still had a muscle pain component.
- “Depending on a case, patients can present either full or partial pain patterns. Δ₸° is the leading SP test factor to determine the number of subjects with muscle-referred pain at a specific ROI.” */* ??? so a single observation would determine a number? So temperature would define pain?
We have simplified the description. The referred pain presence is variable (thigh always, calf not) and Δ₸° is sensitive to the number of subjects with pain in that specific region.
Thanks to your doubts, we have removed all the additional information.
Also, we have noticed that our definition of AURP was not precise enough. We hope we managed to explain it more clearly now.
- “To calculate the significance of the diagnostic parameters in a group of subjects, the thermograms of the examined subgroup were compared every 3 seconds of the test and then the significant differences for every parameter were calculated.” */* ??? I just cannot understand this. NB you do not calculate a significance, you test it. What is a group, a subgroup?
- a) We used Matlab to visualize the differences between the patients for AURP and Δ₸° (separately: data on AURP and data on Δ₸°)
- b) We tested if the differences between the described groups were statistically significant
- c) We checked if the statistically significant difference was directional (higher for GS vs sciatica SP(-); higher for sciatica SP(+) vs sciatica SP (-)) and if we had a significant difference for both diagnostic parameters at the same time. The same was tested for GS vs sciatica SP(+) – to compare the AURP and Δ₸° test results.
It was necessary to show that the observed autonomic phenomenon is not short-lasting (i.e. artifact), but it is an amplified phenomenon increasing with the stimulation time, which slowly decreases during the post-stimulation observation of the patients at rest.
- “Next, the time of the test when both AURP and Δ₸° were confirmed significant was marked as referred pain significant for the specific ROI.” */* “Referred pain significant” has no meaning to me. Neither “the time to the test”. I guess it what the time of occurrence of the first event during the stimulation phase, but I am not sure.
That sentences has been removed.
We have added the following to the SP test description:
Numerical analysis of the thermograms: A comparative analysis of the thermograms (320) registered every 3 seconds and the thermogram recorded for the patient before the stimulation was performed using MATLAB. For every thermogram, the thigh subarea was segmented. Then, both the AURPT0 size and Δ₸° of the thigh region were calculated. The significance of both diagnostic parameters was checked for every out of the 320 thermograms.
- “A simultaneous significant increase or decrease in the two parameters determining the area with amplified vasodilatation or vasoconstriction within trigger-point referred pain provoked during the SP is coincident with the patient’s daily complaint.” */* So there can be decrease/vasoconstriction too? This is new!
Both are possible. We have another paper, showing vasoconstriction, under evaluation. However, vasoconstriction occurs in 5-10 % of cases depending on the muscle.
Other comments:
- This is a test/phenotype relationship study to validate a diagnosis tool.
“2.3 Method”: §1 is actually “Confirmation of phenotypes”.
“2.3.1 Skorupska Protocol” is actually “2.4 Functional exploration”.
What do you mean by “validated”? It seems that here is the validation study.
The validation study requires a different methodology and it was our first study back in 2015. References no 18
Additionally, our method received some awards:
Gold medal with distinction at the International Exhibition of Inventions INPEX 2014 USA for "Skeletal striated muscle examination instrument & new method of myofascial pain diagnosis" (June 2014)
Gold medal at the European Exhibition of Creativity and Innovation EUROINVENT 2014 Romania for "Skeletal striated muscle examination instrument & new method of myofascial pain diagnosis based on the use of the instrument" (May 2014)
Brown medal at 113 International Exhibition of Inventions CONCOURS LEPINE 2014 Paris for "Skeletal striated muscle examination instrument & new method of myofascial pain diagnosis based on the use of the instrument" (May 2014)
Diploma of Polish Minister of Science and Higher Education for the development of the invention: "Skeletal striated muscle examination instrument & new method of myofascial pain diagnosis based on the use of the instrument" (February 2014)
Silver medal at 62 Belgian and International Trade Fair for Technological Innovation Brussels EUREKA 2013 The World Exhibition on Inventions, Research and New Technologies 2013 for "Skeletal striated muscle examination instrument & new method of myofascial pain diagnosis based on the use of the instrument" (November 2013)
Additionally, we would like to emphasize that the innovative nature of our method consists in the observation of amplified vasomotor reactivity suggesting a pathological activity of the autonomic nervous system
A potential role of the autonomic nervous system measurement has been suggested for nociplastic pain. Additionally, trigger points are classified as central sensitization (CS) pain pathomechanism and the presence of referred pain is indicated as a CS feature. The SP test is intended to provoke vasomotor reactivity coincident with muscle referred pain and is recognized as a daily complaint.
Moreover, it is worth mentioning that the applied solutions are a novelty in medical thermovision, particularly as regards the calculation of the percentage size of amplified vasodilatation in addition to the standard temperature measurement. This paper shows for the first time the SP test results for gluteal syndrome and, thanks to the implementation of the Matlab software, it points out some new features of the induced vasomotor response.
- Table 1 (which is a result, not a method): I doubt that the duration of symptoms follow a Gaussian distribution.
We have added the median and Q1/Q3.
- “Skorupska Protocol is a new validated test proposed to objectively confirm nociplastic pain related to muscle.” */* (1) Name it so if you like, but I prefer what I suggested (and probably the journal too); by doing so, Mrs Skorupska will not be able to name so any other test or exploration she develops.
We understand your doubts. We happened to have had similar ones but the reviewers of our papers continued to have reservations about the initial name of the test, which was Minimally Invasive Procedure. That name reminded them of surgical procedures. Another name we considered was Nociceptive Active Dynamic Thermography, which we rejected due to its inaccuracy with respect to, e.g., data analysis. Aware of future limitations, we decided on Skorupska Protocol.
Also, we would like to state that we are planning to perform similar studies on other muscles. Moreover, we would like to check if the size of the observed reaction has clinical meaning, e.g., we would like to evaluate the efficacy of treatment based on the SP test.
We have recently invited new researchers to cooperate with us. Our research is now conducted by an interdisciplinary team including neurologists, orthopedists, pain medicine specialists, musculoskeletal pain specialists, bioengineers, psychologists, neurobiologists, geneticists, and statisticians.
I hope to focus my work on muscle-referred nociplastic pain in the foreseeable future because of the clinical effects I can see in my practice.
- (3) This does not refer to my definition of nociplastic; it looks more like sensitisation, which is nociceptive; fibromyalgia is nociplastic.
- Nociplastic pain is not only fibromyalgia. The newest data indicate that trigger points are recognized as the central sensitization mechanism (nociplastic pain) (Arendt-Nielsen L, et al.. Assessment and manifestation of central sensitisation across different chronic pain conditions.. 2018)(Nijls et al Central sensitization in chronic pain conditions: latest discoveries and their potential for precision medicine; 2021]. Currently, it is indicated that around 30% of patients with musculoskeletal disorders suffer from chronic pain due to the central sensitization process (nociplastic pain). The results of the studies available in the literature were summarized by Harte et al. [Harte et al. The neurobiology of central sensitization. 2018] who proposed two main subtypes of central sensitization, namely “top-down” and “bottom-up”. Fibromyalgia is an example of the top-down CS subtype, and trigger points suit the bottom-up description.
- Statistical methods: there is no Z value (or I missed something).
We have removed the information about the Z value from paragraph 2.5.
- I cannot understand the legends of Fig.1. It is not “visualization”, it is an illustration.
We have corrected that part and added a description to the illustration.
- Reduce the discussion to 1500 words and only focus on
- Summary of the main results
- Comparison to similar studies if any
- Explanations for the results (pathophysiology of GS)
- Biases and limitations
- Consequences for current care or future research
The discussion now follows your recommendations.
- No results in the discussion.
We have removed all the unnecessary information.
NB I will read thoroughly the other tables and the discussion when the rest is addressed.
Kind regards,
authors

This manuscript is a resubmission of an earlier submission. The following is a list of the peer review reports and author responses from that submission.
Round 1
Reviewer 1 Report
As a neuroscientist quite trained in functional exploration in chronic pain patients, I found the manuscript in its present form very hard to follow. I admit that the explorations were very technical – and it may be a part of the novelty – but this does not exempt from a clear explanation of the study objectives. I also wish to remind that any clinical study must identify a primary endpoint (objective) which logically links a pre-study hypothesis and a post-result improvement of the current knowledge, possibly with clinical implications. At the moment, such information is apparently missing.
As a general comment, the manuscript is far too long and the main ideas are not presented as they should, i.e. the main issues, the state of the knowledge, and the primary objective in the introduction, and the definition of groups and possible subgroups in one single chapter of the methods. Scattered bits of information have to be harvested in different parts of the manuscript before being gathered in order to understand, this is harsh. The abstract is to me a good illustration of how confusing is your presentation, I have been unable to understand it.
May be the biggest factor of confusion is your wording of Minimally Invasive Procedure (MIP) and of MIP(+/–) cases. I do not know if this can be changed, but “minimally invasive procedure” is not a specific and informative definition, while you actually conducted a functional exploration of vasomotor reactivity elicited by a noxious remote stimulation. Whether it is minimally invasive or not is just good news for the patient, not a definition of the test. Also, if I understood well, this exploration was firstly dedicated to gluteal syndrome (GS), because pain in the lower limbs is elicited by trigger points located in one of the three gluteal muscles, and that such referred pain is also associated to a vasomotor response in the area of referred pain. If this is right, then all the GS patients have gluteal-evoked referred pain in lower limbs, what you named MIP+ but should therefore be named something like “gluteal trigger”, GT.
This suggests me that the main point (primary objective?) of your study was to explore patients suffering for chronic sciatica (CS) the same way as you currently explore GS, and to compare the vasomotor response of the CS patients, depending on their behaviour GT(+/–), assuming that there could be subgroups of CS with different mechanisms. Again, if I am right, then please state it so very early in the manuscript. In order with this, there must be a clearer definition of the primary outcome, and how you can expect to have enough patients to be able to show a difference between at least the GT+ and the GT– CS patients. I must remind that in such pathophysiological studies, the statistical unit must be the patient and cannot be the test (thermography), because patients can have a different number of ROIs, and this depending on the disease. This does not exclude the possibility of analysing the effect of the ROI on your results (either multivariable analyses adjusted on the ROI, or subgroup analyses by ROI).
You will have to explain better how you defined the GS cases and the CS cases.
- It seems that the diagnosis has been made by different physicians, and this could be a concern for clinician reviewers.
- On which criteria the SLR should clearly differentiate GS and CS?
- Did the GS have a MRI, is it not mandatory to have a negative MRI to define GS?
- You state that CS must have a “lack of trigger points within the gluteus muscles”: this is in contradiction with the further identification of CS-GT+ (so called “MIP+”).
The way you treated the signal is interesting, but this part is too long and technical and should be presented as a supplementary file. Also, the Tables 2 to 5 are unreadable; these results are certainly of interest, but data with 32 repeated measurement times must be presented by curves. There is no need to give the p value for each time, this is anyway a huge type-I error inflation. Better show for each point the 95% confidence interval limits, and make a shift between each group to avoid overlapping. Also remember that you have 3 groups of interest (GS, CS-GT+ and CS-GT–), not 2. Finally, when you compare two groups for repeatedly measured outcomes, you must correct the type-I error inflation, and – as a Bonferroni’s correction would be here awfully conservative (the significance threshold being 0.05/32), I see only two options, either linear mixed models or comparing areas-under-curves.
The recommended plan for a discussion is
- Summary of the main results
- Comparison to similar studies if any
- Explanations for the results
- Biases and limitations
- Consequences for knowledge or future research
The ideal size for and introduction is 500 words, and 1500 for a discussion.
Author Response
Dear Reviewer 1,
Thank you for your time and effort to review our paper. We would like to underline that your comments were the most useful and detailed and they have influenced the final paper the most.
We believe that we have corrected the paper following most of your suggestions. Where needed, the sections have been revised and we added a new subsection titled Consequences for knowledge or future research at the end of the Discussion. Please note that we have decided to summarize all the most important data of both MIP diagnostic parameters in one new table (Table 2) and, thus, removed the previous Tables 2-5.
We would like to explain that our procedure is a new type of active dynamic thermography intended to confirm nociplastic pain related to trigger points. We understand that as an experienced neuroscientist in the chronic pain field you were interested in the autonomic phenomenon induced by the MIP. We agree that its occurrence is crucial because no other type of muscle nociceptive stimulation induced it so far. Other studies examining trigger points or related fields, e.g., acupuncture needle stimulation, have not demonstrated such wide and amplified vasomotor reactivity yet. We believe that in the future the MIP can become an objective tool for the confirmation of nociplastic pain related to trigger points in other diseases. The clinical importance of trigger points related to central sensitization (nociplastic pain) has been indicated in around 30% of different musculoskeletal disorders. That is why our current paper is aimed at both the presentation of the clinical meaning of the MIP are a precise and strict MIP protocol creation.
Thanks to your suggestions, in the Introduction we have added the important information about the MIP being supported by the newly published data, e.g., “The MIP results were considered positive (MIP(+)), if a simultaneous increase in both parameters was confirmed, with ΔTavr as the leading indicator of amplified vasomotor activity. Additionally, according to our previous studies, autonomic phenomenon was confirmed if a specific muscle had active trigger points and the patients reported increased pain sensation during the MIP”.
Furthermore, our newly published paper considered the technical and clinical value of the Matlab involvement in the MIP (Skorupska et al. 2021 Brain Sciences). In the above-mentioned paper, we used a multinomial logistic regression model to analyze both MIP parameters at once (Tavr and AURP segmented picture with Tmax and Tmin being the orientation point for the observation of the isothermal area defined as a vasodilatation or vasoconstriction mark).
The leading meaning of the ΔTavr has been confirmed by the new statistical analysis, according to your suggestion. To obtain the significance of the Z values, a post-hoc Dunn test was performed. Due to the multiple comparison problem, the aforementioned test was corrected by Holm-Sidak procedure. The test was prepared with Dunn test package in R, which again confirmed the primary role of the average temperature changes in defining the autonomic phenomenon presence as significantly valuable. Additionally, please note that we have added information related to the sample size in the paper “Sample size for this publication was limited due to the availability of patients for each group. Therefore, non-parametric tests were performed instead of ANOVA. For the exact test at a significance level = 0.05 and the power of the test beta = 0.95, the lowest possible sample size is N=19. The total sample size for this publication is N=50”. All these data have been added to the paper. However, we considered it important to follow the design of the MIP statistical analysis as presented in all our previously published papers. Tables 2-5 and the Matlab trends are compatible with the data shown in our last study.
We hope we managed to convince you by explaining our point of view. However, we agree that the results could have been unreadable and, thus, to improve their readability we have summarized all the most important data in one table. The rest of the data is now provided as supplementary materials, which can prove important especially to medical thermography specialists.
As regards the clinical value of our study:
As you rightly noticed, we analyzed diseases and patients, not thermograms. There were two kinds of diseases our patients suffered from, namely GS and sciatica. There was no third kind of low back leg pain in our study. What we observed, were atypical reactions to the MIP in the TrPs negative sciatica patients. These data are important for the further development of the MIP protocol and the definition of very precise and strict conditions for objective nociplastic pain diagnosis. This has been already suggested in our newest paper presenting the technical terms of the MIP (Brain Sciences).
Last but not least, we understand that our paper can be at times difficult to follow. First of all, it describes a completely new biological phenomenon, which can possibly become an objective marker of some nociplastic pain subtype. Then, it applies a new type of active dynamic thermography method and for the first time uses invasive nociceptive muscle stimulation. Furthermore, thermogram picture segmentation is performed to confirm pathology. Finally, the analysis of 320 thermograms is conducted using Matlab. We are truly impressed at how your opinion was to the point and so useful that it allowed us to see what important information we failed to include for the reader.
Thank you very much.
We hope you will find our response and the introduced improvements satisfying.
Authors,
Reviewer 2 Report
is important to have more cases in order to not overlook trigger points
Author Response
Dear Reviewer 2,
The paper has been revised according to the suggestions made by Reviewer 1 in order to underline the clinical aspect of the study.
We hope you will find them satisfying. The new version has also been corrected by English Editing Services.
Kind regards,
Authors
Reviewer 3 Report
The study is well designed and presents new information related central sensitization, trigger point and an objective variable like temperature.
I have some comments.
Introduction: One of the most related muscle with gluteal syndrome, in literature, is piriformis’ muscle. Why do not contemplate it? If the reason is that it is a deep muscle which palpation is confusing, this is happen with gluteus minimus too (line 54-55).
Material and methods: ROI appear for first time and there are not a previous explication, we found directly the abbreviation (line 106).
Material and methods: In flow diagram, there are a discordance between assessed to eligibility (n=85) and participants excluded: decline to participate (n=7) and not meeting the inclusion criteria (n=18). 85 – 18 – 7= 60 participants, but only 50 in the groups. What happens with these 10 participants?
King regards
Author Response
Dear Reviewer,
We are extremely grateful for appreciating our work. We have done all mentioned corrections within method section. However, we decided not to change the Introduction part. We would like to explain all our decisions below:
Introduction: One of the most related muscle with gluteal syndrome, in literature, is piriformis’ muscle. Why do not contemplate it? If the reason is that it is a deep muscle which palpation is confusing, this is happening with gluteus minimus too (line 54-55).
The gluteal syndrome has been defined in the ICD-11 for the first time (code XXXI-7) as a Local Syndromes of the Lower Limbs . According to the description the gluteal muscles and quadratus lumborum are considered as a possible pain reason. The piriformis muscle is link to other diseases, which can provoke sciatica like symptoms defined as “Piriformis Syndrome”.
Material and methods: ROI appear for first time and there are not a previous explication, we found directly the abbreviation (line 106).
Thank you very much for your effort and manuscript dissection. We have added the full name before abbreviation “the thigh Region of Interest (ROI)”.
Material and methods: In flow diagram, there are a discordance between assessed to eligibility (n=85) and participants excluded: decline to participate (n=7) and not meeting the inclusion criteria (n=18). 85 – 18 – 7= 60 participants, but only 50 in the groups. What happens with these 10 participants?
Thank you very much for your notification. Due to our inadvertency we wrote (n=7) as the number of participants who decline to participate instead of (n=17). We immediately have corrected our mistake. The new flow diagram has been established and added to the materials.
Yours sincerely,
authors